# Inhibition of Kv10.1 Channels Sensitizes Mitochondria of Cancer Cells to Antimetabolic Agents

**DOI:** 10.3390/cancers12040920

**Published:** 2020-04-09

**Authors:** Ileana Hernández-Reséndiz, David Pacheu-Grau, Araceli Sánchez, Luis A. Pardo

**Affiliations:** 1AG Oncophysiology, Max-Planck Institute for Experimental Medicine, 37075 Göttingen, Germany; ileana.resendiz@gmail.com (I.H.-R.); sanchez@em.mpg.de (A.S.); 2Department of Cellular Biochemistry, University Medical Center Göttingen, 37073 Göttingen, Germany; david.pacheu-grau@med.uni-goettingen.de

**Keywords:** cancer metabolism, drug resistance, mitochondrial dynamics, potassium channel

## Abstract

Reprogramming of energy metabolism constitutes one of the hallmarks of cancer and is, therefore, an emerging therapeutic target. We describe here that the potassium channel Kv10.1, which is frequently overexpressed in primary and metastatic cancer, and has been proposed a therapeutic target, participates in metabolic adaptation of cancer cells through regulation of mitochondrial dynamics. We used biochemical and cell biological techniques, live cell imaging and high-resolution microscopy, among other approaches, to study the impact of Kv10.1 on the regulation of mitochondrial stability. Inhibition of Kv10.1 expression or function led to mitochondrial fragmentation, increase in reactive oxygen species and increased autophagy. Cells with endogenous overexpression of Kv10.1 were also more sensitive to mitochondrial metabolism inhibitors than cells with low expression, indicating that they are more dependent on mitochondrial function. Consistently, a combined therapy using functional monoclonal antibodies for Kv10.1 and mitochondrial metabolism inhibitors resulted in enhanced efficacy of the inhibitors. Our data reveal a new mechanism regulated by Kv10.1 in cancer and a novel strategy to overcome drug resistance in cancers with a high expression of Kv10.1.

## 1. Introduction

During the transformation and progression, cancer cells acquire hallmarks to survive in hostile environments (i.e., low pH, oxygen concentration and nutrients) [1]. Some of the most studied regulators in cancer such as p53, HIF1α (hypoxia inducible factor 1α), c-Myc or K-RAS, play key roles in crucial cell processes like proliferation, cell death, angiogenesis, metastasis, migration or metabolism. 

Since Warburg’s discovery [2], it has become increasingly acknowledged that metabolic changes in cancer cells are crucial for tumor progression and drug resistance [3,4,5]. Cancer cells require a high amount of glucose in comparison with surrounding tissues (20–50 times) [6,7], and this glucose is converted into lactate, even in high oxygen concentrations. Warburg suggested that cancer cells depend exclusively on glycolysis to obtain ATP because the mitochondria are not functional in cancer cells. We now know that mitochondria are not only functional, but rather provide more than 70% of the total energy in a long list of cancer types and is involved in other processes like autophagy, cell cycle regulation, cell death regulation, migration and ROS (reactive oxygen species) homeostasis [8]; in fact, some cancer cells like breast, lung and cervix cancer cells, prefer to obtain energy from the oxidation of amino acids, such as glutamine, rather than from glycolysis [9,10,11]. Nevertheless, it is still accepted that a metabolic rearrangement accompanies malignant transformation. A high glycolytic flux due to an overexpression of glycolytic enzymes regulated by HIF1α, changes in mitochondrial metabolism like a high flux in glutamine oxidation, and optimal capacity of ATP production through the respiratory chain, regulated by p53 and c-Myc [12,13,14], are metabolic pathways that contribute as a strategy for cancer cells to obtain the high amount of energy and de novo molecules needed for anabolic pathways [11]. 

Kv10.1 is a potassium channel ectopically expressed in over 70% human solid tumors from virtually every origin, from sarcoma or melanoma to lung, colon or ovary cancers, and the channel has been proposed as a potential therapeutic target due to the low toxicity of its inhibitors, especially when using antibodies [15]. Recently, we and others have demonstrated that Kv10.1 has a pivotal role in proliferation, angiogenesis, adhesion and migration of cancer cells [16,17,18,19], but neither of those mechanisms alone explains the very high frequency of expression of the channel. Kv10.1 has a close connection with many relevant regulators of cancer cell phenotypes, such as p53, HIF1α and E2F1. Recent evidence indicates that aberrant expression of Kv10.1 can be originated by altered expression of p53, E2F1 and miR-34a [20]. The promoter region of the Kv10.1 gene (named *KCNH1*) displays responsive elements for E2F1 and binding of the transcription factor results in an increased Kv10.1 expression [17], while p53 would maintain low levels of Kv10.1 though the regulation of miR-34a [20]. The overexpression of Kv10.1, in turn, affects cell migration and proliferation through functional interactions with RAB proteins, cortactin and focal adhesion kinase (FAK) [21,22], as well as through calcium signaling [19] and activation of HIF1α [23]

Since p53 is deleted or mutated in most cancer types, resulting in an overexpression of Kv10.1, and p53, E2F1 and miR34a are all regulators of mitochondrial dynamics and function [24,25,26], we investigated the role of Kv10.1 as a regulator of mitochondria. We find that cancer cells expressing the channel become dependent on its expression to maintain their high mitochondrial activity.

Taking advantage of the high energy needs of cancer cells, metabolic inhibitors such as the biguanides metformin and phenformin, or the iron chelator VLX-600 (1-(2-pyridinyl)ethanone(6-methyl-5H-[1,2,4]triazino[5,6-b]indol-3-yl)hydrazine), show promise as therapeutic agents against several cancer entities [27]. Based on our observation that Kv10.1 is required to maintain mitochondrial stability, we tested the possibility that Kv10.1 inhibition increases the efficacy of antimetabolic drugs, and found a strong increase in the potency of mainly phenformin and VLX-600 in several tumor cell models. Our findings open the way for a combined therapy that allows reducing the required dose of antimetabolites and thereby their potential negative effects, such as lactic acidosis. 

## 2. Results

### 2.1. Kv10.1 Regulates Mitochondrial Dynamic Proteins in Cancer Cells

The multiple connections reported between factors relevant for mitochondrial dynamics and regulators of Kv10.1 expression prompted us to analyze mitochondrial dynamics as a process required for the maintenance of proper mitochondrial organization, and for degradation and recycling of damaged mitochondria, thereby influencing both function and morphology of mitochondria. This process is modulated by the cellular energetic state, cell cycle activation and stress environments. Recently, it has been shown that mitochondrial dynamics processes are deregulated in cancer development and play a key role in transformation, growth and metastasis of cancer cells [28,29]. 

To analyze the possible influence of Kv10.1 on this cellular process, we first studied proteins involved in mitochondrial dynamics by Western Blot upon Kv10.1 knockdown. For simplicity, we will refer to cells treated with siRNA as KD (knockdown) cells from here on. Protein extracts from HeLa KD, DU145 KD and their parental control cells were analyzed for the abundance of both fusion (the long isoform of OPA-1 (optic atrophy 1, mitochondrial dynamin-like GTPase), MFN-1 (mitofusin 1) and MFN-2 (mitofusin 2) and fission proteins (DRP-1 (dynamin 1 like), its Ser616-phosphorylated form P-DRP-1, OMA-1 (overlapping activity with M-AA peptidase, metalloendopeptidase) and the short isoform of OPA-1) after 72 h of silencing. HeLa KD and DU145 KD cells showed significantly higher levels of the fission proteins, while a decrease in the content of proteins involved in the fusion of mitochondria (Figure 1a–c) was observed. These results indicate a displacement of the fusion–fission equilibrium towards the latter.

### 2.2. Kv10.1 Knockdown Results in Morphological Fission in Cancer Cells

The increase in the content of proteins involved in mitochondrial fission suggests a change in the mitochondrial morphology in HeLa KD and DU145 KD cells. To determine the activation of fission, the mitochondrial structure in live HeLa and DU145 cells was analyzed by confocal microscopy. Cells were transfected with siRNA and seeded in four-well chambers for microscopy 24 h after transfection. After a further 48 h, the samples were incubated with Mitotracker Deep Red to label mitochondria and Hoechst 33342 (bisbenzimide; Sigma-Aldrich, Munich, Germany) for nuclei and imaged in a spinning disk confocal microscope with environmental control. Confocal images (Figure 2a) showed a high rate of mitochondrial fission in both HeLa KD (Figure 2a,b) and DU145 KD (Figure 2c,d) cells as compared to controls. The degree of mitochondrial fragmentation was quantified by modeling of the mitochondrial network in three-dimensional reconstructions of z-stacks using Imaris software (Oxford Instruments, Abingdon, UK; see example in Appendix A). To improve resolution, we also used SRRF (super-resolution radial fluctuation analysis [30]) and the mitochondrial population in such high-resolution images were analyzed to determine the length of branches in networks [31]. In HeLa KD cells, mitochondria were significantly shorter than in control cells. The images show a clear network in HeLa Control cells by super-resolution analysis (Figure 3a) that suggest fusion/fission dynamicity, while in HeLa KD, the analysis by super-resolution shows network disintegration (Figure 3b). 

To elucidate whether the function of Kv10.1 as a channel is required for its role in mitochondrial dynamics, we used pharmacological blockade of the channel using astemizole, a histamine H1-inhibitor that strongly inhibits Kv10.1, and compared its effects with those of its isomer, norastemizole, which does not block the channel [32]. The cells were treated with the drugs (5 µM) for 24 h, mitochondria were stained as above and their morphology was studied using a spinning disk microscope and SRRF in living cells. Astemizole induced significant mitochondrial fragmentation in HeLa (Figure 4a,e) and DU145 cells (Figure 4c,g).

Astemizole is a hydrophobic drug that acts from the intracellular side of the channel [32]. Therefore, intracellular channels could be implicated in the observed effect. To investigate if plasma membrane channels are required, HeLa and DU145 cells were treated with 10 μg/mL of mAb56, a monoclonal antibody that binds to the extracellular region and specifically blocks Kv10.1. The specificity and IC50 of mAb56 (75nM, approximately 10 µg/mL) has been characterized elsewhere [33]. Control cells treated with mAb56 for 24 h showed fragmented mitochondria in both HeLa (Figure 4b,f) and DU145 (Figure 4d,h), to an extent significantly larger than cells treated with a control antibody which binds to a neighboring region of the channel but is incapable of inhibiting its function (mAb62, [33]) suggesting that the presence of active Kv10.1 at the plasma membrane is necessary to maintain mitochondrial network formation. Similar results were obtained in DU145 cells treated with antibody, although DU145 cells show already mitochondrial fragmentation under control conditions. Taken together, these results show a direct correlation between Kv10.1 function and mitochondrial fusion/fission regulation and correlate with the high content of proteins involved in fission processes observed in Western Blots. 

### 2.3. Activation of Fission by Kv10.1 Depletion Results in Enhanced Autophagy

Mitochondrial dynamics process can be activated through different signaling pathways; a high content of ROS and the accumulation of damaged mitochondria can activate fission processes in order to decrease oxidative stress and eliminate damaged mitochondria [34,35]. The activation of fission is the first step for mitophagy, the form of autophagy (a process to remove unfolded proteins and clear damaged organelles) that eliminates mitochondria when required [28]. To determine if the activation of fission in KD cells results in enhanced autophagy, the content of proteins involved in this process (Phosphoinositide 3-kinase class III, PI3KC3; and microtubule-associated protein 1A/1B-light chain 3, LC3B) were analyzed in HeLa KD and DU145 KD cells. In the absence of Kv10.1, the content of autophagy markers was 2–5 times higher than in control cells (Figure 5a–d). The results correlate with an increase in the content of lysosomes in HeLa KD cells imaged by super-resolution microscopy (Figure 5e), indicating activation of mitophagy that matches the enhancement of fission when Kv10.1 is absent, suggesting a correlation between Kv10.1 activity and mitochondria stability. 

### 2.4. Activation of Autophagy Correlates with Low Mitochondrial Protein Content

Until now, our results suggest that the absence of Kv10.1 can induce the activation of mitochondrial fission and mitochondrial digestion through autophagy. In order to analyze if autophagy activation results in a decrease of mitochondrial proteins, we determined by Western blotting the levels of proteins involved in oxidative phosphorylation (ND1 (complex I), Rieske (complex III), COX 4–1 (complex IV), ATP5B (complex V)), at different time points after transfection (24, 48, 60 and 72 h), to measure a possible decrease in mitochondrial proteins as a result of mitochondrial degradation in HeLa KD (Figure 6a,c) and Du145 KD (Figure 6b,d). In both cell lines, mitochondrial proteins showed a significant decrease (40–80%) as compared to control cells; the effect was minimal 24 h after transfection, and became more evident thereafter, in good agreement with the time required for Kv10.1 knockdown by siRNA. These results strengthen the hypothesis that the absence of Kv10.1 causes activation of mitochondrial fission and autophagy and, resulting in a lower content of mitochondrial proteins in HeLa KD and Du145 KD. Therefore, we conclude that mitochondrial stability is compromised in KD cells. 

### 2.5. Mitochondria from HeLa KD and Du145 KD Show a Decrease in Transmembrane Potential (ΔΨm)

Fragmented mitochondria typically display altered mitochondrial membrane potential (ΔΨm) [36]. This phenomenon could be a consequence or a cause for altered levels of mitochondrial proteins as mitophagy is triggered to remove damaged mitochondria. To explore the interconnection of Kv10.1 expression with mitochondrial fitness, we analyzed the impact of the absence of Kv10.1 on ΔΨm in HeLa and DU145 cells. KD and control cells were treated with rhodamine 6G (R6G), a positive lipophilic compound that is attracted by the negative potential in the inner mitochondria [37]. The distribution of R6G intensity was analyzed by flow cytometry in 10,000 cells per experiment. HeLa KD and Du145 KD cells showed a significant decrease of R6G internalization as compared to controls (Figure 7), suggesting that the mitochondria of cells with lower Kv10.1 expression have lower membrane potential, or that alternatively, KD cells have fewer mitochondria, which would correlate with the decrease in mitochondrial proteins and activation of autophagy described above. 

### 2.6. Fission Activation in KD Cells Correlates with Reactive Oxygen Species (ROS) Enhancement 

The results obtained from autophagy and mitochondrial potential analysis suggest that Kv10.1 is needed for mitochondrial stability and recycling, and that absence of Kv10.1 results in damaged mitochondria. It has been shown that fission activation can be linked to the production of reactive oxygen species (ROS) [35]. We thus measured ROS content in cells with endogenous overexpression of Kv10.1 upon gene silencing. HeLa cells were treated with 10 μM 2′7′-Dichlorofluorescein diacetate (DCFHDA) after 72 h of silencing. Fluorescence was measured using an IncuCyte (Sartorius, Göttingen, Germany) live cell imaging system. Cells treated with 10 μM rotenone, 100 μM H_2_O_2_, or 30 μM resveratrol were used as positive (rotenone, H_2_O_2_) or negative controls (Figure 8a). The quantification of fluorescence showed a 10-fold enhancement in DCFHDA signal in HeLa KD cells vs. control cells (transfected with a control siRNA; Figure 8b), suggesting that the absence of Kv10.1 compromises the cellular redox state resulting in an overproduction of ROS and activation of mitochondrial fission and autophagy). 

To determine whether a decrease of surface expression or the requirement of a functional Kv10.1 is responsible for this observation, non-silenced cells were incubated for 12 h with 10 μg/mL of the blocking antibody mAb56. The use of mAb56 in control cells also produced a modest increase in the DCFHDA signal, which was not significant when normalized for cell confluence. 

### 2.7. Cell Lines with High Levels of Kv10.1 Have a High Sensitivity to Mitochondrial Inhibitors

Our data strongly suggest that Kv10.1 promotes mitochondria stability in cancer cells. Therefore, targeting the channel in tumor cells could enhance the efficacy of mitochondrial inhibitors to treat cancers overexpressing Kv10.1.

To test this hypothesis, we determined the IC50 for growth inhibition by metformin, phenformin and VLX600 in eight cell lines with different expression levels of Kv10.1 (Figure 9a). Interestingly, growth inhibition was markedly stronger in cell lines with robust Kv10.1 expression (HeLa, DU145, MDA-MB-435 and SHY5SY) than in those with low expression (Panc-1, MiaPaCa2 and MDA-MB-231cells). Importantly, the sensitivity to metabolic drugs is not shared by non-tumor cells. When non-cancer hTERT-RPE1 cells were treated for 48 h with metformin, phenformin and VLX600, no toxic effects where observed at the concentrations used for cancer cells, suggesting that these metabolic drugs show higher efficacy in malignant cells and are even more effective in cells with high expression of Kv10.1 (Appendix A).

Similar results were obtained for phenformin and VLX-600 when cytotoxicity was measured by live cell imaging in IncuCyte using a cell death marker (Cytotox Green). The toxicity was 2-8-fold higher in HeLa and DU145 cells treated with phenformin and VLX600 than in MiaPaca2, Panc-1 and MDA-MB-231 (Figure 9b). Metformin was the drug showing weakest toxic effects in all of the cell lines studied, although it was able to arrest proliferation (Figure 8b). Plotting the normalized toxicity versus the abundance of transcripts as measured by RT-PCR, a linear correlation between Kv10.1 expression and cytotoxicity of all three drugs was observed, with the sole exception of MDA-MB 231 cells treated with VLX-600 (Figure 9c). This correlation is specific for Kv10.1 (Appendix A); the sensitivity to metabolic drugs was independent of the expression levels of Kv10.2 or Kv11.1, both closely related voltage-gated potassium channels. 

Since the metabolic drugs used in the previous experiments are not specific for mitochondria [38], it is not clear if the sensitivity for metabolic drugs is due to mitochondrial damage or other metabolic pathways are involved in the observed response. To determine if the mitochondria are involved in this sensitivity, we determined the IC50 value for rotenone, a specific inhibitor for complex I of the mitochondrial respiratory chain, on the growth of HeLa, DU145, MDA-MB-435, MCF-7 and MiaPaca2 for 12 and 24 h of exposition. Rotenone was more efficient inhibiting the growth of cells with high expression levels of Kv10.1 (Figure 9d). These results correlate with the mitochondrial sensitivity that HeLa KD and Du145 KD show through the activation of mitochondrial degradation processes.

In a similar set of experiments as described above, HeLa KD and Du145 KD and their parental control cells were treated with metformin, phenformin and VLX600 for 48 h and their proliferation was assayed by continuous imaging. All drugs were more effective in KD cells (Figure 7e), with IC50 values 40% lower than those of control cells. Therefore, knockdown of Kv10.1 in HeLa and DU145 further increased the sensitivity for metabolic drugs, indicating that the higher sensitivity of cells expressing Kv10.1 is a reflection of higher dependence on mitochondria in those cells, rather than a direct implication of the channel in the effect of the drugs. 

### 2.8. Blockade of Kv10.1 Enhances the Cytotoxic Effect of Metabolic Inhibitors

We then set out to determine if the inhibition of Kv10.1 produces a similar effect as knockdown on the sensitivity to metabolic drugs. For the design of combined therapy, we used mAb56, which is able to arrest the growth of cancer cells through its inhibitory action on Kv10.1 [33], together with the metabolic drugs used above. We combined the antibody (10 μg/mL) with sub-IC50 (for KD cells) concentrations of metformin (5 mM), phenformin (100 μM) and VLX600 (0.1 μM), and followed the growth of HeLa for 48 h using live cell imaging (Figure 10a,b). Treatment with mAb56 inhibited cell growth in the presence of metabolic inhibitors in an additive fashion. In contrast, ROS production was enhanced synergistically by the combination of phenformin and Kv10.1 blockade in HeLa cells (Figure 10c). In control cells, mAb56 alone or phenformin had no significant effect. Combination of mAb56 with phenformin or VLX600 induced significantly increased ROS production. In contrast, in the absence of Kv10.1 (HeLa KD), there was an increase in ROS production under all conditions with respect to control cells. ROS production was further enhanced by phenformin and VLX600. As expected, in the absence of Kv10.1, mAb56 treatment produced no additional changes. 

## 3. Discussion

In all tissues and cell types, the production of ATP, second messengers ((Nicotinamide adenine dinucleotide phosphate, NADPH; inorganic phosphate, Pi; guanosine diphosphate, etc.) and the synthesis of de novo molecules are pivotal processes to maintain growth and cell survival. In particular, cancer cells have a high demand for energy and molecules like proteins and DNA due to the high rate of proliferation, movement (migration and metastasis), angiogenesis and cell death resistant processes. Cancer metabolism has often been proposed as a good target for cancer treatment. Crucial transcription factors such as p53 or HIF-1α can regulate mitochondria homeostasis in cancer cells [39,40] and control the expression of enzymes involved in glycolysis and mitochondrial metabolism (glucose transporter, hexokinase II, 6-phosphofructokinase, pyruvate kinase, lactate dehydrogenase, cytochrome c oxidase subunit IV, glutaminaseetc.). In general, HIF-1α regulates negatively the mitochondrial metabolism while p53 is a positive regulator. It has been shown that HIF1α is overexpressed in some cancer types, especially in tissues with low oxygen concentration where it increases the glycolytic rate and the production of lactate. On the other side, almost 70% of all cancer types have mutations or absence of p53, resulting in its loss of activity as a positive regulator of mitochondria, so the idea that p53 regulates mitochondria stability in cancer cells is questionable due to the absence of the protein or mutations that this gene presents in many types of cancer. Here, we propose Kv10.1 as a novel mitochondrial regulator in cancer cells through the maintenance of mitochondrial networks.

Evidence that potassium channels are major regulators in cancer development has grown dramatically in the last decade. Dysregulated expression of potassium channels in many human cancers results in increased metastatic potential and proliferation and correlates with poor prognosis in cancer patients. In our group, we investigate Kv10.1, in which expression is found in >70% of all types of cancer and participates in the regulation of key processes in transformation and tumor growth. Besides its role Kv10.1 in cell cycle regulation and primary cilia disassembly, Kv10.1 has a close relationship with HIF1α and p53 pathways. p53 represses Kv10.1 [20] and Kv10.1 increases HIF1α expression [23]. 

In normal and cancer cells, proper regulation of the recycling of damaged/old organelles is required to prevent cellular stress and oxidized protein accumulation. Mitochondrial hyperfission causes high contents of ROS and loss of mitochondrial potential and oxygen consumption. The mitochondrial elongation is activated in high-energy demand state to protect the cell from oxidative stress and cell death activation [41]. Seemingly, Kv10.1 is needed to maintain mitochondrial fusion and network formation. We hypothesize that since cells with high levels of Kv10.1 have a high rate of proliferation, they need a high amount of energy and de novo molecules. A coupled mitochondrial network is important to maintain the proliferation rate, low ROS and ATP demand. The experiments using mitochondrial inhibitors demonstrate that cell lines with overexpression of Kv10.1 show higher sensitivity than cells with low expression of Kv10.1, suggesting a dependence of mitochondria to obtain the necessary to survive, and the blockage of Kv10.1 result in a loss in mitochondrial stability and an even more remarkable sensitivity to mitochondrial drugs. 

The exact mechanism of how Kv10.1 stabilizes mitochondria is not yet fully understood. The fact that a non-permeable blocker, such as mAb56, recapitulates the effects of loss of Kv10.1 indicates that permeation through the plasma membrane channels is required for this function of Kv10.1. Nevertheless, non-canonical functions of ion channels independent of ion permeation have been also documented (e.g., see [42]). At this point, we cannot rule out that ion flow through Kv10.1 is only a part of the signaling mechanism. Active Kv10.1 channels participate in primary cilium disassembly [16,43], a process that is also impaired by mitochondrial stress [44]. Both ciliary homeostasis [45] and mitochondrial dynamics [46] require calcium signaling, and active potassium channels hyperpolarize the membrane, increasing the driving force for calcium to enter the cells. In addition, potassium homeostasis on its own influences mitochondrial function [46]. It has been shown that Kv10.1 interacts with the calcium entry channel Orai1 in some cancer cell types [19,47]. In immune cells, calcium entry through Orai1 induces repositioning of the mitochondria to influence calcium signaling [48,49,50]. It is tempting to speculate that dysregulation of calcium homeostasis and its interplay with the mitochondria [51] and the primary cilium is at the basis of the mitochondrial impairment that we report here. 

## 4. Materials and Methods 

### 4.1. Cell Culture 

The human cell lines: DU145 (DSMZ ACC 261), MDAMB231 (DSMZ ACC 732), Panc-1 (DSMZ ACC 783) and hTERT RPE1 (ATCC CRL-4000) were grown in Dulbecco’s MEM medium supplemented with 10% fetal bovine serum. SHYS5Y (DSMZ ACC 209), HeLa (DSMZ ACC 57) and MCF-7 (DSMZ ACC 115) were grown in RPMI medium supplemented with 10% fetal bovine serum, MDAMB435S (ATCC HTB-129) were grown in RPMI medium supplemented with 15% fetal bovine serum and MiaPaca-2 (DSMZ ACC 733) cells were grown in DMEM-F12 medium supplemented with 10% fetal bovine serum. All media were purchased from Thermo Fisher Scientific (Schwerte, Germany), and bovine serum from Biochrom (Berlin, Germany) or PAA (GE Healthcare, Freiburg, Germany). All cell lines were incubated under a humidified atmosphere of 95% air/5% CO_2_ at 37 °C. 

### 4.2. siRNA Transfection

For knockdown experiments, we used INTERFERin (Polyplus, Illkirch, France)) or Lipofectamine RNAiMAX (Invitrogen, Thermo Fisher Scientific, Schwerte, Germany)) according to the respective manufacturer protocol. Briefly, HeLa (3 × 105 cells), DU145 (5 × 105 cells), MCF7 (5 × 105 cells) and SH-SY5Y (7 × 105 cells) were seeded in T75 flasks (Greiner, Frickenhausen, Germany) using the corresponding medium, and allowed to adhere for 12 h. Then cells were transfected with a validated Kv10.1 siRNA sequence: CAGCCAUCUUGGUCCCUUATT (Qiagen, Hilden, Germany) or with a siRNA control sequence (Ambion, Thermo Fisher Scientific, Schwerte, Germany) at 30 nM final concentration in Opti-MEM (Gibco, Thermo Fisher Scientific, Schwerte, Germany)) medium during 4 h. After transfection, the Opti-MEM medium was replaced with the corresponding supplemented medium. Unless otherwise indicated, cells were harvested 48 h after transfection for the experiments. After 72 h, the health of the cells did not allow us to draw conclusions. The siRNA sequence has been extensively validated elsewhere [52], and the extent of knockdown under the present experimental conditions was validated by Western blot (Appendix A).

### 4.3. Western Blot 

Changes in protein content were determined through Western Blot assays. Cells were collected by trypsinization and cell pellets were mixed and incubated at 4 °C with 200–250 μL of lysis buffer (20 mM Tris-HCl, 5 mM EDTA pH 8, 150 mM NaCl, 1% Nonidet P-40, 0.5% sodium deoxycholate, 0.1% SDS) for 30 min and then centrifuged at 10,000 × *g* for 20 min. Protein concentration was determined by BCA (bicinchonic acid) assay, and proteins were subsequently separated by polyacrylamide gel electrophoresis for 45 min at 150 V. After separation, the proteins were transferred to nitrocellulose membranes using wet transfer (BioRad, Feldkirchen, Germany; 12 mM Tris-Base, 96 mM Glycine, 20% methanol) for 2 h at 50 V. The membranes were rinsed twice with distilled water and treated with Pierce Western Blot Signal Enhancer according to the manufacturer’s instructions. Membranes were incubated with the primary antibody of interest at the dilutions indicated by the manufacturer. The corresponding secondary antibodies (anti-mouse or anti-rabbit, Amersham, GE Healthcare, Freiburg, Germany) were used at 1:7000 dilution. Densitometry analysis was performed by using the Image J software [53] in FIJI [54].

### 4.4. Determination of Mitochondrial Transmembrane Potential

Mitochondrial membrane potential (ΔΨm) was determined by flow cytometry. Cancer cells were seeded at 5 × 105 per well in 6-well plates and treated with 0.1 µM rhodamine 6G (Sigma, Darmstadt, Germany) for 20 min at 37 °C. Cells were resuspended in 1 mL PBS and analyzed using a BD FACS Aria (Becton Dickinson, Ashland, OR, USA). Mitochondrial membrane potential was determined comparing cells treated with 5 μM carbonyl cyanide 3-chlorophenylhydrazone (CCCP) (Sigma Darmstadt, Germany)) as a negative control. The fluorescence analysis was performed with FlowJo software v. 10 (Becton Dickinson, Ashland, OR, USA). 

### 4.5. Mitochondrial Shape and Mitophagy Determination

Mitophagy was assessed by detecting the mitochondria and/or lysosomes by confocal microscopy. Afterwards, 3 × 104 cells were cultured in μ-Slide 4 well sterile slides (Ibidi, Gräfelfing, Germany). Forty-eight hours after transfection, cells were treated with 10 μg/mL Hoechst 33342 (nuclei) (Sigma-Aldrich, Munich, Germany), 100 nM MitoTracker Deep Red (mitochondria) (Thermo Fisher Scientific, Freiburg, Germany) and 250 nM LysoTracker Red DND-99 (lysosomes) (Thermo Fisher Scientific, Freiburg, Germany) for 15 min at 37 °C. Z-stack images were collected using a 100× objective (Nikon Apo TIRF 100× Oil, NA 1.49; Nikon Instruments, Amsterdam, Netherlands) with a Nikon-Andor spinning disk confocal microscope (Andor iXon Ultra 888 camera; Oxford Instruments, Abingdon, UK) using Zeiss F2 oil. Images were analyzed in FIJI. Stacks of 100 images for super-resolution radial fluctuation analysis (SRRF) were acquired at 100Hz and processed in FIJI using the NanoSRRF plugin [30]. The resulting images were analyzed for mitochondrial morphology also in FIJI [31] and the mitochondrial length was determined in the resulting skeleton. Three-dimensional reconstruction and analysis of the mitochondrial network were performed using Imaris software (v. 9.3.0, Oxford Instruments, Abingdon, UK). Alternatively, high-resolution images were obtained in a Zeiss LSM 880 microscope equipped with an Airy detector and processed using Zeiss Zen software (Oberkochen, Germany).

### 4.6. Proliferation, Cytotoxicity and ROS Detection by Live Cell Imaging

The different cell lines were seeded in 96-well plates (3–5 × 103) (Greiner, Frickenhausen, Germany) in fresh media. After 12 h, cells were treated with the relevant concentration of the different drugs to obtain the IC50 on proliferation and cytotoxicity: metformin (2.5, 5, 10, 15, 20 and 25 mM), phenformin (50, 70, 100, 250, 500 and 1000 μM), VLX600 (0.1, 0.25, 0.5, 0.75, 1 and 2.5 μM) and rotenone (0.5, 1, 2.5, 5 μM) in quadruplicates. The wells were then imaged at 2h intervals (three images per well) for 48 h using an IncuCyte device (Sartorius, Göttingen, Germany). Proliferation was measured by percent confluency in phase contrast, and cytotoxicity was assessed using Cytotox Green (Sartorius, Göttingen, germany) following the protocol recommended by the manufacturer. Cytotox Green is a membrane-impermeant dye that fluoresces upon DNA binding and, therefore, stains dead cells. The IC50 on proliferation was determined by non-linear regression with GraphPad (San Diego, CA, USA) Prism 5–8. For ROS detection, cells were treated with 100 μM 2′7′-Dichlorofluorescin diacetate (DCFHDA) for 15 min. After treatment, cells were washed with PBS and DCFHDA fluorescence was determined as above.

## 5. Conclusions

We show that Kv10.1 expressed on the surface of cancer cells exerts protective actions on mitochondria, which can be a part of the selective advantage conferred to the cells that explains the remarkable high frequency of overexpression of Kv10.1 in tumors. This advantage is, however, at the cost of increasing the dependence of the cell of mitochondrial metabolism, which renders the cell more sensitive to metabolic drugs. This opens the possibility of combinational therapies simultaneously targeting the mitochondria and the channel, which could improve efficacy and reduce the side effects of the individual therapeutic agents.

## Figures and Tables

**Figure 1 cancers-12-00920-f001:**
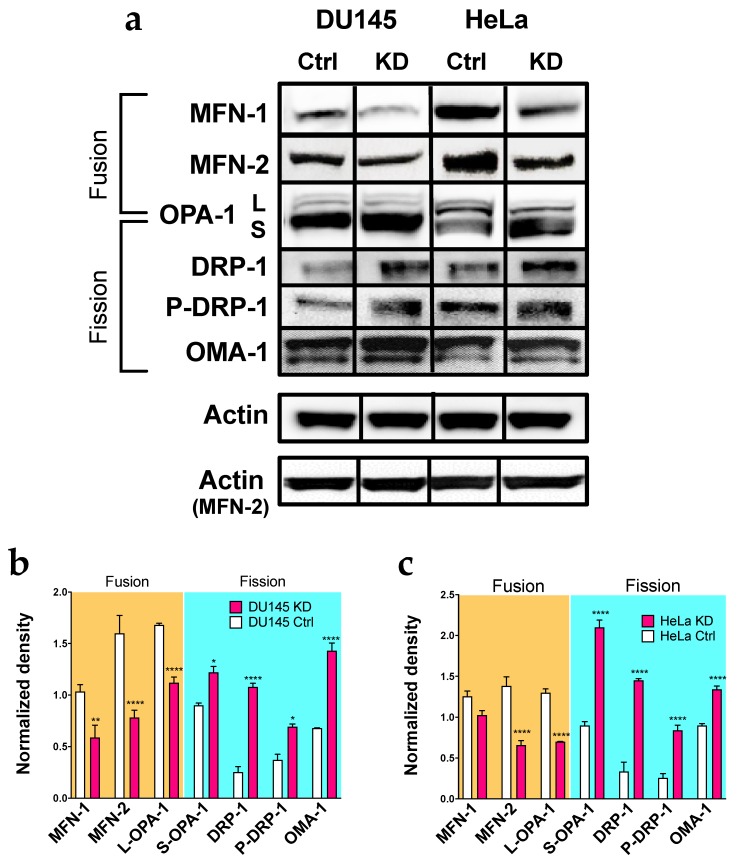
Silencing for Kv10.1 results in an overexpression of proteins involved in fussion processes. (**a**) representative images of Western Blot of proteins involved in mitochondrial dynamics in DU145, DU145KD, HeLa and HeLa KD cells. Actin was used as a loading control in all cells. The bands correspond to t = 72 h of the same blot for each cell line, except for MFN-2, which is a different membrane, and therefore has a different actin loading control. (**b**,**c**) Densitometric analysis of the protein content in DU145, Du145KD (**b**), HeLa and HeLa KD cells (**c**); the values were normalized to the loading control. Bars represent the mean of the three independent experiments. Error bars indicates SEM (* *p* ˂ 0.05, ** *p* ˂ 0.001, **** *p* < 0.0001, 2-waz ANOVA).

**Figure 2 cancers-12-00920-f002:**
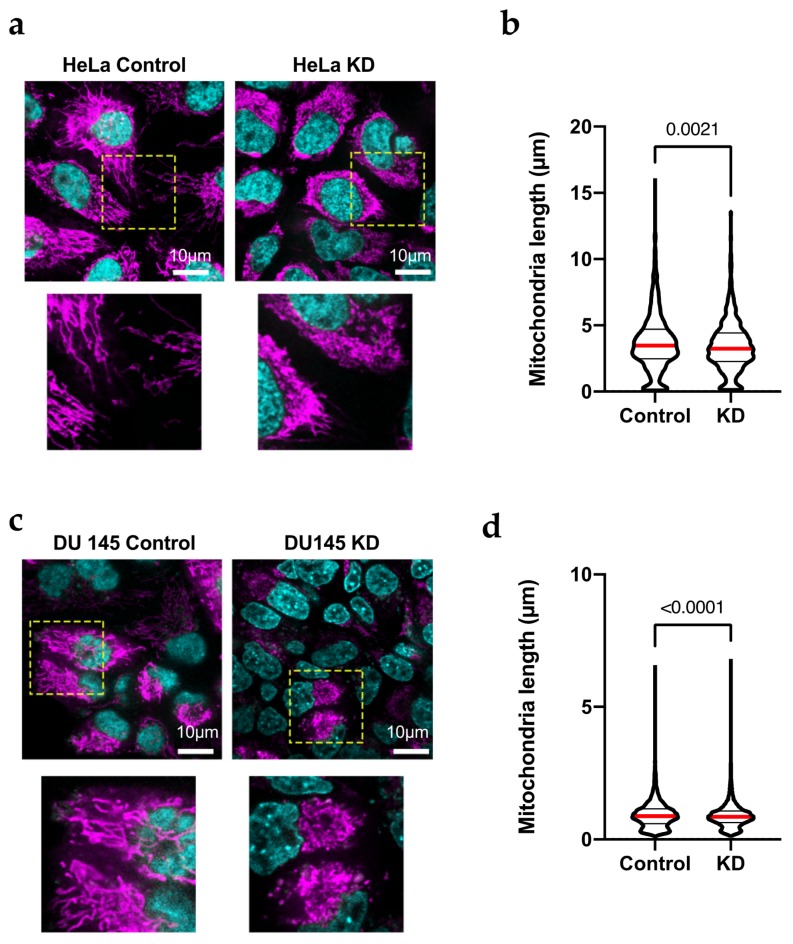
HeLa KD cells show mitochondrial fragmentation. (**a**,**c**) Confocal images of HeLa (**a**) and Du145 (**c**) cells transfected with siRNA against Kv10.1 for 48 h, stained with Mitotracker Red (magenta, mitochondria) and Hoechst 33342 (czan, nuclei) and analyzed by confocal microscopy. Mitochondria show fragmentation in KD cells while control cells show elongated mitochondria. The are inside the yellow square is shown magnified below. (**b**,**d**) The length of individual mitochondria in HeLa and Du145 cells was determined by filament tracking analysis of 3-dimentional reconstructions using Imaris software. In both cases, KD cells showed shorter mitochondria, In b and d, the median value is indicated by a red line and the number indicates the value of p obtained by non-parametric Mann-Whitney test since the limit in resolution renders the distributions not normal.

**Figure 3 cancers-12-00920-f003:**
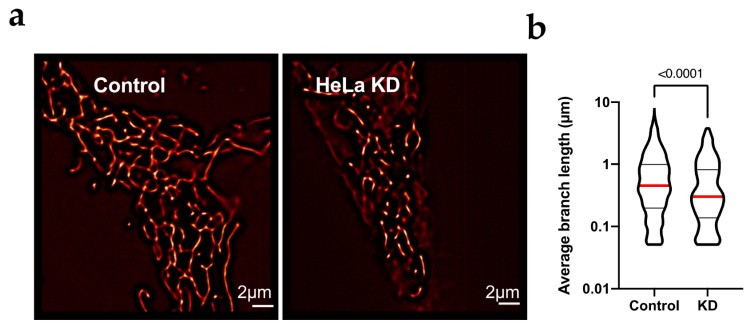
HeLa KD cells show mitochondrial fragmentation. (**a**) For detailed structure analysis, stacks of 100 images were analyzed by SRFF in HeLa and HeLa KD cells; representative examples are shown. (**b**) Average branch length in HeLa cells was larger then in HeLa KD cells. The median value is indicated by a red line and the number indicates the value of p obtained by non-parametric Mann-Whitney test since the limit in resolution renders the distributions not normal.

**Figure 4 cancers-12-00920-f004:**
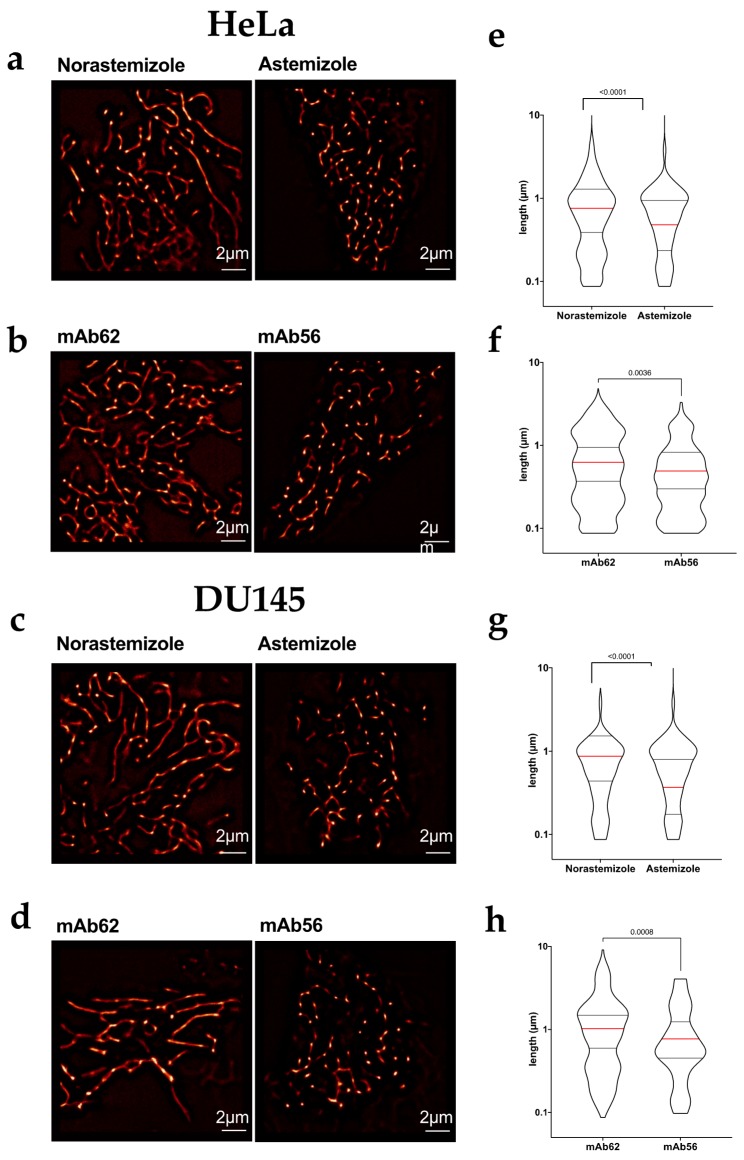
Pharmacological blockage of Kv10.1 induces mitochondrial fragmentation (**a**–**d**) Representative SRRF images of HeLa (**a**,**b**) and Du145 (**c**,**d**) cells treated with the indicated agents for 24 h, stained with Mitotracker Red and imaged in vivo using stacks of 200 images by spinning disc microscopy. Both blockers, Astemizole (5 µM) and mAb56 (10 µg/mL) induced mitochondrial fragmentation as compared with the respective controls Norastemizole and mAb62 at the same concentration. (**e**–**h**) The length of individual mitochondria under each condition was estimated by identifying skeletons in Fuji software. In e–h, the median value is indicated by a red line and the number indicates the value of p obtained by non-parametric Mann-Whitney test since the limit in resolution renders the distributions not normal.

**Figure 5 cancers-12-00920-f005:**
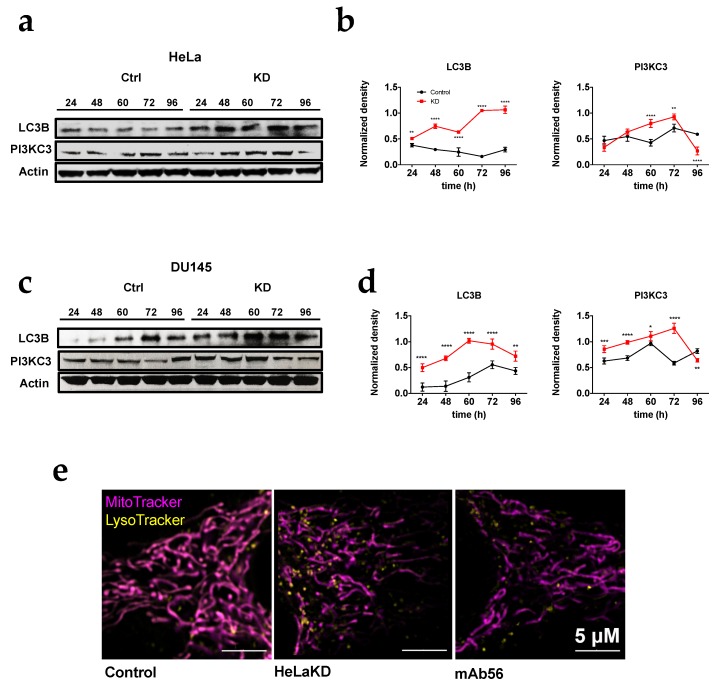
HeLa KD and DU145KD cells show an increase in autophagic components. (**a**) Representative blots of proteins involved in autophagy in HeLa and HeLa KD cells. Actin was used as a loading control in all cases. (**b**) Densitometric analysis of the autophagic protein content HeLa and HeLa KD cells the values were normalized to actin. (**c**) Representative Western blots in DU145 and DU145KD cells. (**d**) Densitometric analysis of the autophagic protein content DU145 and DU145KD cells the values were normalized to actin. (**e**) Cells treated with control siRNA (left), Kv10.1 siRNA (center) or mAb56 (right) were incubated with Mitotacker (mitochondria, magenta) and Lysotracker (lysosomes, yellow) and analyzed in live cells by confocal microscopy (Airy super resolution). Note the increase in LysoTracker signal in KD and mAb56 treated cells. values in (**b**,**d**) are mean ± SEM of three independent experiments (* *p* < 0.05. ** *p* < 0.01. **** *p* < 0.001, **** *p* < 0.0001, 2-way ANOVA).

**Figure 6 cancers-12-00920-f006:**
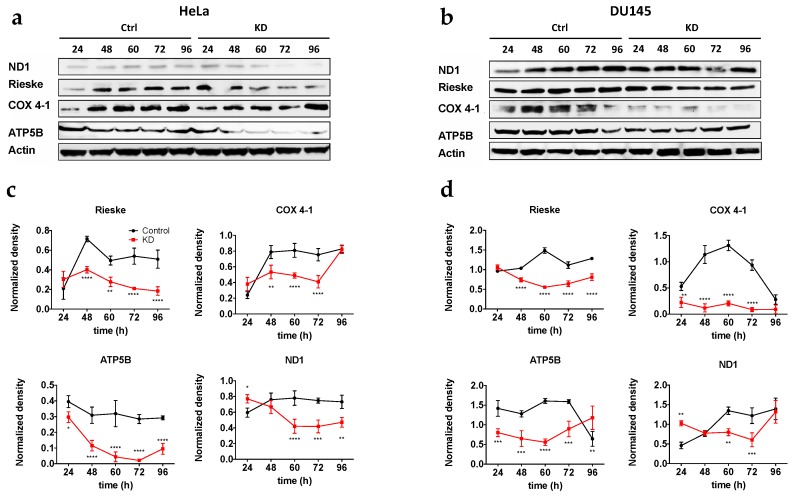
Knockdown of Kv10.1 results in a decrease in mitochondrial proteins. Representative blots of HeLa and HeLa KD cells (**a**) and DU145 and Du145KD cells (**b**). Actin was used as a loading control in all cells. (**c**,**d**) Densitometric analysis of three independent experiments in HeLa (c) and Du145 (d) cells. Values were normalized to actin (Mean ± SEM * *p* < 0.05. ** *p* < 0.01. **** *p* < 0.001, **** *p* < 0.0001, 2-way ANOVA).

**Figure 7 cancers-12-00920-f007:**
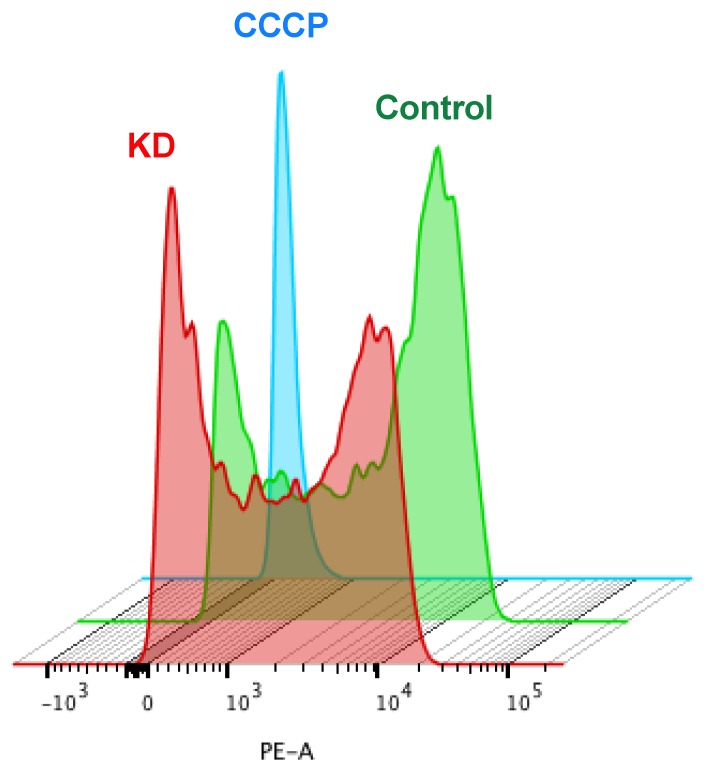
The absence of Kv10.1 results in a decrease in mitochondria potential. HeLa cells were transfected with siRNA for Kv10.1 and 48 h later treated with Rhodamin 6G. The mitochondrial membrane potential was measured using flow cytometry. The histogram presented is representative of three independent experiments. KD cells (red) were more depolarized then controls (green). The blue histogram represents cells whose mitochondria was depolarized by treatment with CCCP.

**Figure 8 cancers-12-00920-f008:**
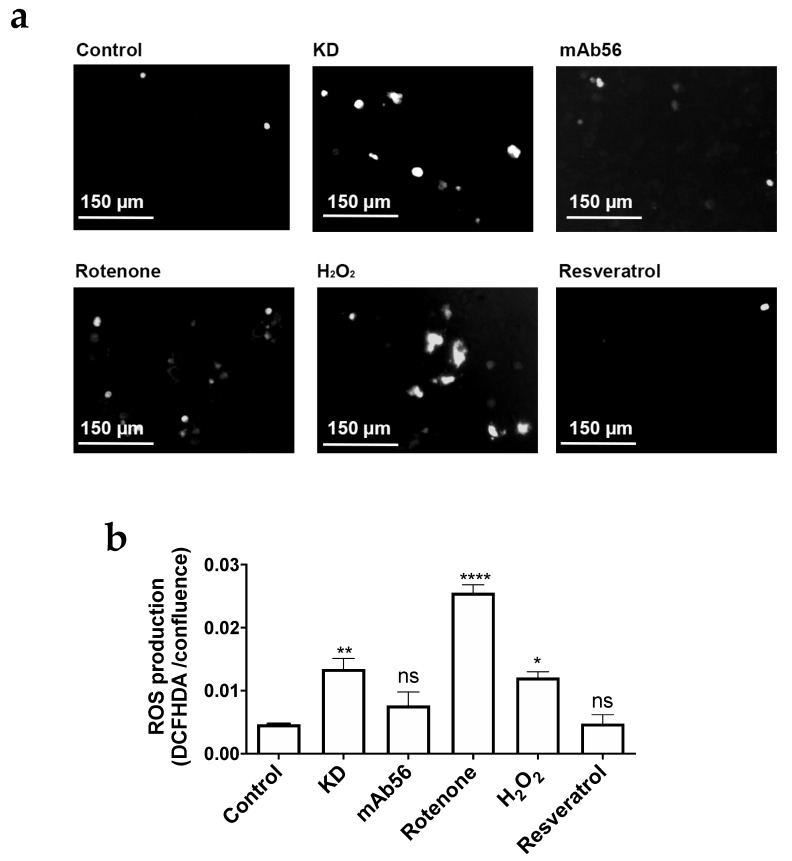
KD cells show n increase in ROS. (**a**) Representative images of the different treatments. HeLa cells were seeded in 96 well plates and stained with DCFHDA. The green fluorescence of DCFHDA was used as a reporter of ROS production using Incucyte device. The different conditions were Kv10.1 knockdown (72 h), mAb56 (10 mg/mL, 12 h), rotenone (10 µM, 12 h), H2O2 (100 µM, 20 min) and resveratrol (30 µM, 24 h). (**b**) Quantification of ROS production (expressed as integrated fluorescence intensity normalized for cell confluence in phase contrast). Mean ± SEM of three independent experiments; * *p* < 0.05; ** *p* < 0.01; **** *p* < 0.0001, ANOVA.

**Figure 9 cancers-12-00920-f009:**
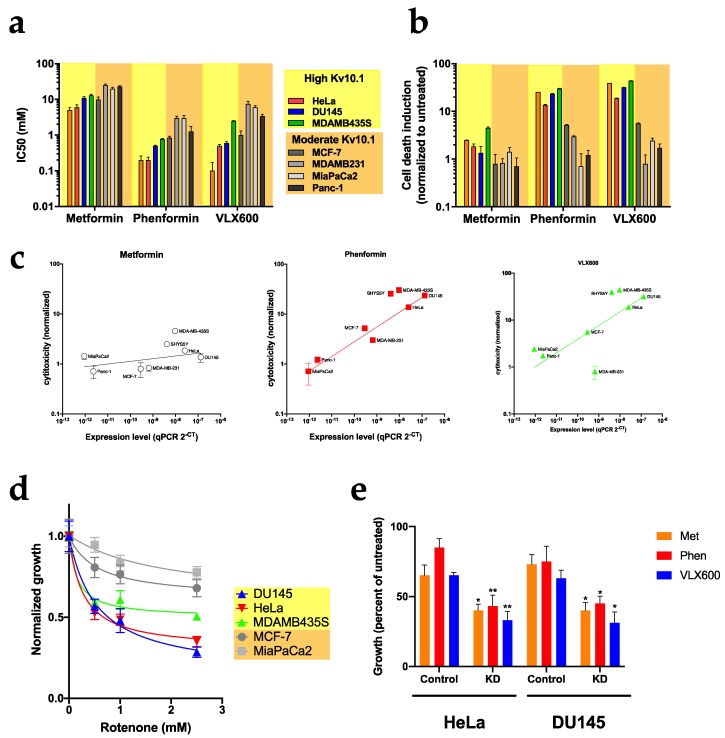
Cell lines with overexpression of Kv10.1 show high sensitivity to metabolic drugs. (**a**) IC50 of the metabolic drugs on the growth of the indicated cell lines measured as confluence using phase contrast in a Incucyte live cell imaging system. (**b**) Cellular toxicity in the different cell lines treated with metabolic inhibitors. Cells were labeled with Cytogreen to detect death cells. (**c**) Correlation of cytotoxicity with expression level of Kv10.1 in the different cell lines. The normalized cytotoxicity was plotted against the threshold values for Kv10.1 in RT-PCR. (**d**) The growth of the indicated cell lines in the presence of different concentrations of rotenone was determined and is presented normalized to the growth in untreated cells. Cell lines with high Kv10.1 expression were more sensitive than those with low levels. (**e**) HeLa and DU145 cells were transfected with siRNA for Kv10.1 for 48 h and treated with the IC50 of the different metabolic inhibitors. Bars represent the mean ± SEM of three independent experiments (* *p* < 0.05; ** *p* < 0.01, two-way ANOVA).

**Figure 10 cancers-12-00920-f010:**
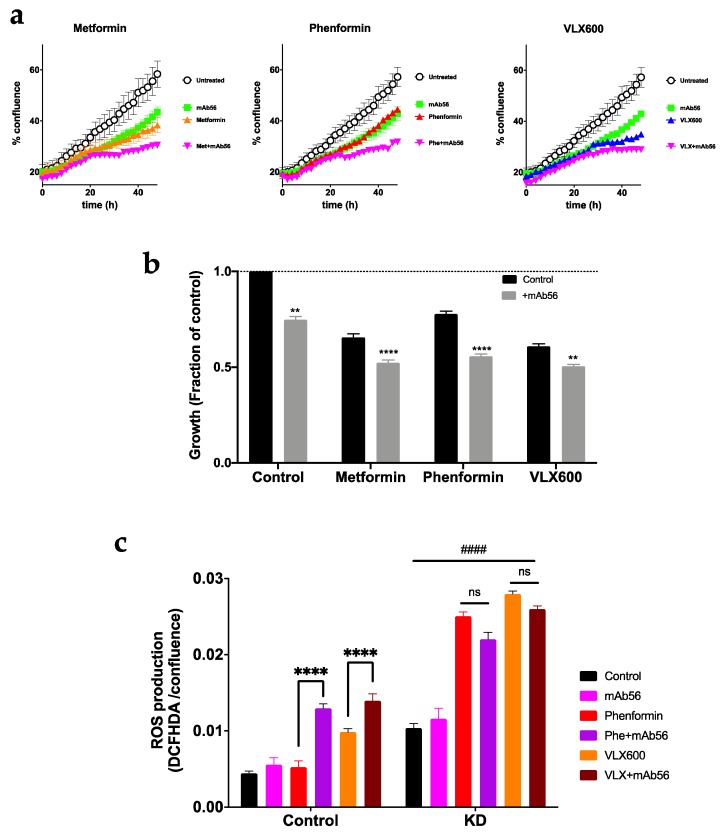
Combined treatment using metabolic inhibitors and mAb56 show higher efficacy than monotherapy. (**a**) Growth of HeLa cells treated with metformin (5 mM), phenoformin (100 µM) and VLX600 (100 nM) alone or in combination with an antibody specific for Kv10.1, mAb56 (10 µg/mL). Confluence was measured by phase contrast every two hours for 48 h. (**b**) Comparison of growth at the endpoint of the experiment. (**c**) ROS production was measured after 48 h of treatment using DCFHDA fluorescence. The values were normalized by the confluence in each well, as an estimation of the total number of cells. Bars represent the mean ± SEM of three independent experiments (** *p* < 0.01; **** *p* < 0.0001, two-way ANOVA). In C, #### indicates *p* < 0.0001 for each condition compared to the respective controls.

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
