# Peer review of "Inhibition of Kv10.1 Channels Sensitizes Mitochondria of Cancer Cells to Antimetabolic Agents"

_cancers, 2020, doi:10.3390/cancers12040920_

Round 1

Reviewer 1 Report

The authors addressed most of my concerns and I support the publication of the manuscript.

Reviewer 2 Report

I am still not satisfied with readability of old Fig. 3 / new Fig. 4 (fonts in part too small) but this is only of minor issue and does not affect content or soundness of the MS.

Good work!

This manuscript is a resubmission of an earlier submission. The following is a list of the peer review reports and author responses from that submission.

Round 1

Reviewer 1 Report

In general, the manuscript is written properly presenting a coherent  and logic narrative.

Altogether the data support the notion of authors that the activity of KV10.1 affects dynamics of mitochondria and knock-down of Kv10.1 results in mitochondria fragmentation.  

However, there are several complementary experiments which in my mind should be provided to support obtained data.

Major points:

1) The level of Kv10.1 protein by Westen blot and mRNA by RT-PCR in comparison to the level in the control cells should be shown to support knocked-down (KD) cell line phenotypes. In this regard, it would be also appropriate to use mAb56 antibody to show its specificity.

2) Precise quantification of protein content based on Western blot experiments is very challenging and usually exhibits high variability.  Since the data on the level of mitochondrial fusion/fission proteins are the starting point for the rest of the experiments, therefore the data should be confirmed by independent methods i.e. RT-PCR.

It would be also appropriate to provide for review all the western blots that were used for quantification of the protein content.

Minor points:

1) Figure 3 a,c and Figure 4 a, b -  units (h) should be indicated on the picture for description of the numbers 24, 48, 60..... even though it could be inferred from the text.

line 38 -  "We now know that mitochondria is not only functional , but rather 38 provides more than 70% of the total energy in a long list of cancer types and is involved in..."- should be are and provide since mitochondrion sing.  as opposed to mitochondria pl.

line72 -  "of–mainly–phenformin" should be "of mainly pherformin"

Author Response

1)The level of Kv10.1 protein by Westen blot and mRNA by RT-PCR in comparison to the level in the control cells should be shown to support knocked-down (KD) cell line phenotypes. In this regard, it would be also appropriate to use mAb56 antibody to show its specificity.

We thank the reviewer for his/her comment. We had omitted mentioning that control, but it had been performed. We have included the data as new Figure S3. We did not perform protein determinations in every case in the presence of antibody because the volumes used in those experiments would require very large amounts of antibody. Nevertheless, regarding the specificity of mAb56, we have now been more explicit and included some more details, while maintained the reference to the original detailed description of the antibody.

2) Precise quantification of protein content based on Western blot experiments is very challenging and usually exhibits high variability.  Since the data on the level of mitochondrial fusion/fission proteins are the starting point for the rest of the experiments, therefore the data should be confirmed by independent methods i.e. RT-PCR.

We agree with the reviewer that western blotting is a rather qualitative method. Nevertheless, RT-PCR or other methods share many of the limitations (i.e., a change in RNA does not always linearly reflect on protein abundance and/or activity). We would not suggest that there is a change in protein abundance based only on western blots, but the fact that we see a functional correlate complements and reinforces the western blot data

3) It would be also appropriate to provide for review all the western blots that were used for quantification of the protein content.

We have provided the information as required by the Journal.

Minor points.

We have corrected those mistakes, thank you.

Reviewer 2 Report

I have no suggestions for changes.  However, the manuscript mentions supplemental materials and the link is given on line 363.  However, that link is not functional, and I was therefore unable to access the supplemental materials.  

Author Response

I have no suggestions for changes.  However, the manuscript mentions supplemental materials and the link is given on line 363.  However, that link is not functional, and I was therefore unable to access the supplemental materials.

Thank you very much. We apologize for this inconvenience. To avoid such problems, the supplementary material is now in the same file.

Reviewer 3 Report

The authors aim to address the impact of Kv10.1/EAG1/KCNH1 on mitochondrial stability and regulation by several assays/methods. Based on WB-detection of proteins involved in mitochondrial Fusion or Fission, EAG KD favors mitochondrial Fission. Mitochondrial fragmentation is accompanied by a reduction in branch length (Confocal fluorescence images) and followed by autophagy. This consistently results in a decrease in mitochondrial proteins involved in oxidative phosphorylation (WB data), a decrease of mitochondrial potential (R6G-based FACS analysis), and an increased ROS level in KD cells (DCFHDA fluorescence). Tumor Cell lines with increased EAG expression on the one hand seem to be more sensitive towards mitochondrial inhibitors (IncuCyte microscopic analysis). On the other hand also knockout of EAG in HeLa and DU145 cells renders them more sensitive towards mitochondrial inhibitors. Co-application of EAG-inhibiting mAb56 may increase the antiproliferative efficiency of mitochondrial inhibitors, thus representing a putative treatment strategy against (EAG-expressing) cancer.

By this the authors provide an impressive data set on the impact of EAG1 channels in the plasma membrane on mitochondrial regulation, function and impact on cellular state and proliferation. However, they leave an important question unanswered: Is the ion permeating function involved in all (there may be multiple) signals arising from EAG1 channels to affect mitochondria?

Off note: Data provided in this manuscript are not consistent, since mAb56 application in one experiment (Fig. 6b) increased ROS production as efficiently as siRNA-mediated KD, whereas in the other experiment (Fig. 8c) KD, but not mAb56, was effective.

Therefore, to address this question, additional experiments have to be performed to unambiguously clarify whether ion permeation is required for signaling from EAG1 to mitochondria. For this, the effects of mAb56 should be compared with those of mAb62 (a non-inhibiting antibody also binding in the outer pore region) and other EAG1-inhibiting LMW compounds imipramine, clofilium/LY97241, haloperidol (they do not have to distinguish between EAG1 and ERG1, since the latter appear not to be involved (Fig.  S3B)). Ion permeation may be required/important for certain signaling function from EAG1 to mitochondria, but also an internalization and redirection of antibody-bound EAG1 to internal membranes may be feasible.

Moreover, some improvements on the manuscript itself are necessary:

WB data that show the time course of EAG1 knockdown are required to interpret functional effects of the knockdown itself and to allow a comparison with data arised from EAG1 inhibition by mAb56. Here it is also of interest of whether mAb56, at the concentration used, is expected to inhibit EAG1 by more than 50%.

What exactly is shown in Fig. 3e? Labels are missing and legend is not sufficient.

Legend of Fig. 6 also has to be extensively improved.

Minor:

Fig. 1: As judged by eye, the strength of some bands (HeLa - MFN1, DU145 – OPA1) do not appear to be compatible with the mean data

What is P-DRP-1? Phosphorylated DRP-1? And if so, phosphorylated at which site / by which enzyme?

Fig. 2: Why are data for DU145 (as mentioned in the text) not shown?

Figs. 3+4: Why are densitometric data shown for only up to 72 h, whereas WB data go up to 96 h?

Fig. 4a+b: Positioning of labels are not optimal

Fig. 5: Compared to data from e.g. ref #35, the number of cells with depolarized mitochondrial membrane potential seems extremely high (or is this (Y) log scale?

May the reduced fluorescence result from less mitochondria with unchanged potential, so was this possibly misinterpreted?

Fig. 6: What about data for DU145, as mentioned in the text?

Fig. S3: here it would be interesting to know, whether hTERT-RPE1 cells express Kv10.1 and – if not – whether transfection with Kv10.1 renders the cells susceptible towards mitochondrial inhibitors

Fig.7 Cytogreen principle/method is not explained

How were data on rotenone effect generated (IncuCyte/confluence measurements?)

Why were cell lines MDAMB231 and Panc-1 not included ?

Panel 7e: Why are control data clearly above 50% although inhibitors are applied at IC50 concentrations?

Fig. 8: mAb56 by itself decreases proliferation of HeLa cells. From my point of view, there is no evidence for an increased efficiency of the mitochondrial inhibitors - only n sum mAb56 and the inhibitors are more effective.

Again, Du145 data are mentioned in the text but not shown in the figure

Author Response

By this the authors provide an impressive data set on the impact of EAG1 channels in the plasma membrane on mitochondrial regulation, function and impact on cellular state and proliferation. However, they leave an important question unanswered: Is the ion permeating function involved in all (there may be multiple) signals arising from EAG1 channels to affect mitochondria?

We thank the reviewer for her/his encouraging comments. We agree that this is an important questions and have performed an additional set of experiments that provide evidence for the requirement of active plasma membrane channels for the effects observed. Nevertheless, this does not rule out other signals through the channel, and we have highlighted this possibility in the discussion.

Off note: Data provided in this manuscript are not consistent, since mAb56 application in one experiment (Fig. 6b) increased ROS production as efficiently as siRNA-mediated KD, whereas in the other experiment (Fig. 8c) KD, but not mAb56, was effective.

We thank the reviewer for pointing this out and apologize for this error. The data presented in Figure 6 (now Figure 7) corresponded to raw florescence data, and were not corrected for cell density. We have now corrected this mistake. There is still a tendency to increase in ROS with mAb56, which we attribute to the acute application of the antibody. We deem this observation interesting, but we have not pursued it further.

Therefore, to address this question, additional experiments have to be performed to unambiguously clarify whether ion permeation is required for signaling from EAG1 to mitochondria. For this, the effects of mAb56 should be compared with those of mAb62 (a non-inhibiting antibody also binding in the outer pore region) and other EAG1-inhibiting LMW compounds imipramine, clofilium/LY97241, haloperidol (they do not have to distinguish between EAG1 and ERG1, since the latter appear not to be involved (Fig.  S3B)). Ion permeation may be required/important for certain signaling function from EAG1 to mitochondria, but also an internalization and redirection of antibody-bound EAG1 to internal membranes may be feasible.

We thank the reviewer for this suggestion. We agree that this is an important information and have followed the advice. The additional experiments (summarized in the new Figure 3) examined the effect of mAb62 and that of the LMW drug astemizole, an efficient blocker of Kv10.1 and Kv11.1, but that offers the advantage that its isomer norastemizole is unable to block channels while retaining the primary targets of astemizole. The results are compatible with the need of inhibition of the current at the plasma membrane to induce mitochondrial fragmentation.

WB data that show the time course of EAG1 knockdown are required to interpret functional effects of the knockdown itself and to allow a comparison with data arised from EAG1 inhibition by mAb56. Here it is also of interest of whether mAb56, at the concentration used, is expected to inhibit EAG1 by more than 50%.

Please see the response to Reviewer #1. The time course is now included as Figure S3. We have also added information on the IC50 of mAb56, which is close to the 10 µg/ml we use. This is however just an estimation based on short-term experiments in transfected cells and performed in saline solution at room temperature.

What exactly is shown in Fig. 3e? Labels are missing and legend is not sufficient.

Thank you for pointing this out. We have added the labels and improved the legend.

Legend of Fig. 6 also has to be extensively improved.

We have corrected this

Minor:

Fig. 1: As judged by eye, the strength of some bands (HeLa - MFN1, DU145 – OPA1) do not appear to be compatible with the mean data

The image presented is a representative blot of three-four different experiments. We have exchanged the image for one where the difference is more clear, but we also carefully revised the analysis to discard any mistakes and confirmed our previous values. Because the blots presented do not correspond to the same membrane, we have removed the actin lane.

What is P-DRP-1? Phosphorylated DRP-1? And if so, phosphorylated at which site / by which enzyme?

As the reviewer guesses, P-DRP1 stands for phosphorylated. CDK1/cyclinB or CDK5 phosphorylate DRP-1 in Ser 616 for mitochondria fission activation The antibody used is specific for phosphorylation in Ser616 (Taguchi N., Ishihara N., Jofuku A., Oka T., Mihara K. (2007).J. Biol. Chem. 282, 11521–11529. 10.1074/jbc.m607279200; Liesa M., Palacín M., Zorzano A. (2009). Physiol. Rev. 89, 799–845. 10.1152/physrev.00030.2008). I

Fig. 2: Why are data for DU145 (as mentioned in the text) not shown?

They were included as supplemental data that apparently was not accessible for the reviewers. The new Figure shows also this cell line.

Figs. 3+4: Why are densitometric data shown for only up to 72 h, whereas WB data go up to 96 h?

We decided to plot only the time course until 72h because the health of the cells was compromised after that time, and the control cells also had significantly reduced levels of the channel. We include now a new a Figure including the 96h time point (also in the analysis, which introduces a slight quantitative variation in p values)

Fig. 4a+b: Positioning of labels are not optimal

We have corrected this

Fig. 5: Compared to data from e.g. ref #35, the number of cells with depolarized mitochondrial membrane potential seems extremely high (or is this (Y) log scale?

The Y axis is normalized to show all three histograms with the same height. The distribution of CCCP-treated cells is much narrower than that of non-treated cells, and that gives the impression of many more depolarized cells. Ref #35 describes a different cell type, and as it is probably known to the reviewer, the appearance of flow cytometry results depend a lot on the initial selection of “viable cells”.

May the reduced fluorescence result from less mitochondria with unchanged potential, so was this possibly misinterpreted?

We thank the reviewer for this observation, this is indeed a possibility and we have mentioned it in the new version.

Fig. 6: What about data for DU145, as mentioned in the text?

We thank the reviewer for pointing this out. ROS determination were performed only in HeLa cells. We have corrected the text accordingly.

Fig. S3: here it would be interesting to know, whether hTERT-RPE1 cells express Kv10.1 and – if not – whether transfection with Kv10.1 renders the cells susceptible towards mitochondrial inhibitors

hTERT-RPE1 cells doe express Kv10.1, although in a time restricted manner, as described in reference #16 (Sánchez et al., 2016).

Fig.7 Cytogreen principle/method is not explained

Thank you for pointing this out. We have corrected this.

How were data on rotenone effect generated (IncuCyte/confluence measurements?)

We have described this in more detail in the Methods section and the Figure legend.

Why were cell lines MDAMB231 and Panc-1 not included ?

We limited the screen to three cell lines attending to sensitivity to metabolic inhibitors (low, MiaPaCa2, intermediate, MCF7 and high, MDA MB 435S), and added Hela and Du 145 because we had studied the changes in dynamicity in those cell lines.

Panel 7e: Why are control data clearly above 50% although inhibitors are applied at IC50 concentrations?

The IC50 concentration used for the experiments in 7e (now 8e)correspond to the IC50 for KD cells;since Ctrl cells are more resistant to metabolic inhibitors (fig. 7a and 7b), they show more than 50% inhibition. We have clarified this in the text.

Fig. 8: mAb56 by itself decreases proliferation of HeLa cells. From my point of view, there is no evidence for an increased efficiency of the mitochondrial inhibitors - only n sum mAb56 and the inhibitors are more effective.

We agree with the reviewer in the case of proliferation, and we state this clearly now in the text. However, we do see sensitization in ROS production. We have now explicitly clarified this.

Again, Du145 data are mentioned in the text but not shown in the figure

As mentioned above, DU 145 cells were not tested for ROS production, but in this case we believe the text was correct.

Round 2

Reviewer 1 Report

There are several important problems with the revised version of the manuscript.

The main issue is concerning the presentation, reproducibility and statistical significance of Western blot data.

Authors answered in the comments to my review: " We would not suggest that there is a change in protein abundance based only on western blots, but the fact that we see a functional correlate complement and reinforces the western blot data". However, they start their story with Western blot data and these data prompted them for further investigation.  If these data are not fully convincing they should not be included in the main manuscript or added just as supplementary data.

First, only one set of Western blots was provided for revision even though statistical data are based on three/four repetitions.

Second, no loading controls were provided at all for the Western blots in the supplementary data set. As a result, it is impossible to judge if the statistics are consistent with the raw data. For instance, there is a high variability of the signal on Western blot with anti-OPA-1 antibody (supplementary data set, Ctr1 to Ctr4), which is inconsistent with small error bars in Fig.1c. Is this due to uneven gel loading?

Third, some Western blots reveal poor specificity of antibodies (e.g. Beclin1). Are authors sure about the identity of the band that they quantify? Is the molecular mass the sole criteria (by the way, why there are now protein mass markers shown)?  In these cases, antibodies should be validated with KD of protein of interest.

Fourth, some Western blots are of poor quality. For instance, Western blot for Beclin seems to have a weaker signal on the left side than on the right (bands and background), the one for Rieske has air bubble artifacts that interfere with quantified bands.

In Figure 1 a,  each lane looks like it comes from one gel but instead it is a patchwork from different blots and positions in one blot. It should be clearly indicated by separators. This figure without loading controls for each band makes little sense. Here also header present in the previous version of the figure is missing.

In summary, Western blot data are not convincing at all at this point and I suggest focusing on fewer proteins (or which most specific antibodies are available) with fewer time points and do convincing quantification (with serial dilution, for instance, see reference https://www.hindawi.com/journals/bmri/2019/5214821/ ).

I cannot find captions for Figures  2, 3, 5, 6, 7!! and I cannot see also Figure S1, which makes it more difficult to interpret data. 

However, I have questions here too. In Figure 2b, the length distribution and median length between HeLa Control and KD cells are very similar. Was the number of cells or mitochondria counted the same between these two groups? Why different statistics (sum of mitochondria length) was shown for the second cell line (DU145). Fig 2a and 2c are of low resolution/detail and do not contribute to the presentation of the data.

In Figure 5, no raw data for COX4-1 were provided.

In Figure 7b, the authors change the ROS production quantification to account for cell density. Therefore, it would be appropriate to add corresponding phase-contrast images to Figure 7a.

Reviewer 3 Report

The manuscript improved substantially.

Unfortunately, also in the improved 2nd version, most figure legends are missing. Hence, I cannot assess changes/improvements made here. But I may believe…

Minor changes/suggestions:

In figure 1 the labeling of the diff. lanes (DU145 Ctrl/KD ; HeLa Ctrl/KD) may be added (as in the original version)

The authors may double-check all references to (new) figure (panels). E.g. line 122:     (Fig. 3 a, e) and DU 145 cells (Fig. 3 c,g)   instead of    (Fig. 3 a, b) and DU 145 cells (Fig. 3 e,f).

To increase readability fonts in Fig. 3 (violin plots) and 8c may be increased

Still, line 246 claims HeLa and DU145 cells were investigated, but Fig 9 shows data for HeLa only

Spelling of DU145 may be harmonized (in part Du145)